# Digital Workflow in Full Mouth Rehabilitation with Immediate Loading, Intraoral Welding and 3D-Printed Reconstructions in a Periodontal Patient: A Case Report

**DOI:** 10.3390/reports6040052

**Published:** 2023-11-01

**Authors:** Adam Nowicki, Karolina Osypko

**Affiliations:** 1Diamante Clinica Dental Clinic, Sportowa 48 A/C, 59-300 Lubin, Poland; adamusnowicki@gmail.com; 2Platinum Clinic, Księcia Witolda 49, 50-202 Wrocław, Poland

**Keywords:** computer-aided implantology, computer-guided surgery, immediate loading, intraoral welding, 3D printing, periodontitis, case report

## Abstract

Background: Complex implant reconstructions in patients with residual dentition due to periodontitis is a challenging task in many aspects. Methods: This study shows a full digital workflow combining 3D printing, guided implant placement, intraoral scanning and welding with immediate loading and digital smile design. An analog impression was taken to validate the passive fit of final restorations. The whole treatment plan was divided into three stages. The first stage included an intraoral scan of baseline dentition, and then the extraction of all teeth was performed, implanting four temporary implants and providing the patient with removable temporary prosthesis. The second stage was to scan the removable temporaries, implanting 10 implants and multi-unit abutments (MUA), and create a rigid construction via the intraoral welding of titanium bar and by fixing it to the 3D-printed temporary reconstructions (designed with DSD) as a form of immediate loading. The third stage included the scanning of screw-retained temporary reconstructions, then scanning from the MUA level and creating final reconstruction. Results: The presented workflow enabled the delivery of some sort of restoration to the patient at every moment of the treatment and to sustain the required esthetic effect with decent comfort of use even in the early stages. Conclusions: A full digital workflow is a reliable treatment method even in complex cases.

## 1. Introduction

Successful extensive rehabilitations of patients with periodontal disease are usually jeopardized by the scale and complexity of necessary steps and unforeseen factors. Clinicians need to connect surgical, periodontological and prosthetic knowledge to acquire satisfying results and always insure patients with some sort of temporary restoration for healing periods in between subsequent stages of the treatment plan. In such cases, a digital workflow combined with newest technological advances may become a helpful way to fulfill this goal and enable patients to regain their smiles as soon as possible. This case represents a well-fitted combination of different aspects of digital dentistry from digital smile design, the 3D printing of surgical guides and suprastructure shell, intraoral welding and finally CAD-CAM designs of final restoration.

## 2. Case Report Section

### 2.1. Patient Information and Clinical Findings

A 75 y.o. patient made an appointment at the dental office with complaints of halitosis, a lack of function and abnormal mobility of their remaining teeth that led to articulation and mastication malfunction (Figure 1 and Figure 2A,B). Regarding the social activity of our patient and their professional commitment as an academic tutor, there was no possibility of even temporary edentulism. According to the patient’s medical history, the patient had no chronic diseases and has not taken any medicines on a daily basis.

The initial idea of immediate implantation after the extraction of remaining teeth was discarded, as according to studies such as [1] there is no sufficient data on the survival rate of immediate loaded implants after extractions performed because of aggressive periodontitis (GAP) [1]. According to a recent classification, this can be perceived as periodontitis grade IV stage C [2].

Therefore, we was decided to conduct the following:The extraction of remaining teeth and immediately give removable 3D printed dentures inserted on the temporary implants to the patient;Implantation after the healing time and the immediate loading of temporaries;Final restoration after another six months.

This workflow was chosen as a method of higher predictability [3,4,5].

### 2.2. Therapeutic Intervention and Outcomes

#### 2.2.1. Stage I

The fully digital workflow was applied with the use of scanners and 3D printers [6]. At first, an intraoral scan was taken (IO scan) (Figure 3) with the use of Medit i500 (Medit Corporation, Seoul, Republic of Korea). As the patient had no particular esthetic demands at this point of treatment, a digital smile design protocol and face scan were not needed at that point in order to restore function and desirable esthetics. Removable dentures were designed based on the intraoral scan, just after digital tooth extraction (Meshmixer, Autodesk, Inc., ver:3.5.474, San Francisco, CA, USA). They were manufactured via 3D printing using DLP technology (Phrozen mini 4k, Phrozen Tech. Co., Ltd., Hsinchu City, Taiwan) with the following NextDent resins: a light pink base and C&B bleach (NextDent B.V., Soesterberg, The Netherlands).

On the day of surgery, after multiple extractions, tooth sockets were carefully curetted and filled with leukocyte platelet-rich fibrin (L-PRF, Intra-Lock, Boca Raton, FL, USA). Afterwards, four temporary implants (two in the maxilla and two in the mandible) were implanted without a flap, into the interdental septa, without any surgical guide and with manual positioning (Figure 4A,B). They were implanted in positions 14 and 24 (FDI Dental Numbering System) in the maxilla, and in positions 34 and 43 in the mandible. All of these procedures were conducted with the use of local articaine + adrenaline anesthesia (40 mg + 0.01 mg)/mL). Temporary machined implants were characterized by a smooth surface, dimensions of 2.9 mm in width and 10 mm in length, and a tip with a ball-head attachment (ICX-mini, Medentis medical, Bad Neuenahr-Ahrweiler, Germany). These implants are sufficient for temporary, mucosa-supported immediate restoration during the healing phase of permanent implants. Matrices for ball-head abutments were placed in 3D-printed temporaries in order to achieve proper retention (Figure 5A,B). Due to the healing process and possible prosthetic field aberrations, teeth were blocked with flat cusps, while group guidance and three-point occlusal contact in both arches were chosen (Figure 6A,B).

After the procedure, the patient was prescribed an antibiotic: amoxicillin with clavulanic acid in an amount of 875 + 125 mg to be taken every 12 h for 7 days. No premedication was given to the patient.

#### 2.2.2. Stage II

After the healing process, which in the discussed case took four months, the removable dentures were realigned with impression material (Variotime light body, Kulzer, Mitsui Chemical Group, Düsseldorf, Germany). Hot gutta-percha, which is usually used for root canal treatment (Endopilot, Schlumbohm GmbH & Co. KG, Brokstedt, Germany), was utilized as a radiological marker, after being applied on the labial and buccal surfaces of the dentures (Figure 7).

To maintain the proper and repeatable occlusal position, translucent and non-radio shading material was used (Harvard TransMatrix, Harvard Dental International GmbH, Hoppegarten, Germany). In the normal resting position of the jaw, CBCT and an I.O. scan (Intra Oral scan) with an additional 360° denture scan were performed (Figure 7). With the help of the radiological markers, the I.O. scans were merged with CBCT (Blue Sky Bio, LLC, Libertyville, IL, USA) (Figure 8). Additional feedback from the patient resulted in the conduction of additional digital smile design (DSD) due to a canted occlusal plane (Smile Designer Pro, Tasty Tech LTD., Toronto, ON, Canada). Based on DSD, a new tooth chain was designed, which helped to ensure the proper position of implants in the alveolar bone and to make them more vertically loaded. Then, surgical guides for implant placement were designed with temporary implants and pins as a means for their improved stabilization (Figure 9A–C).

Implants (ROOTT, Trate Ag, Root, Switzerland) were placed in both arches with local anesthesia articaine + adrenaline (40 mg + 0.01 mg)/mL). In the maxilla, six implants were implanted vertically. In the mandible, two implants in the anterior region were implanted vertically and the other two marginal implants were angulated. Primary stability reached 40 Ncm [7] and multiunit abutments (MUA) were screwed into the implant platforms. The temporary implants were screwed out manually, without a flap. Within the one-day-surgery philosophy, the loading of the implants was carried out with the AM abutments screwed on top of the MUA, which was intraorally welded with 2.0 titanium wire to obtain a rigid and passive framework (Figure 10). The next step was immersing the rigid frameworks into the 3D hollow printed shells that had balanced occlusion [8], as the removable prosthesis was a biocopy of an existing prosthetic field (the residual dentition). The fixation between the rigid framework and printed shells was made with LuxaPick-up composite (DMG Chemisch-Pharmazeutische Fabrik GmbH, Hamburg, Germany) and Composite Primer by GC (GC International AG, Luzern, Switzerland). These procedures allowed the creation of long-lasting temporaries that met the function and esthetic demands for the time of osseointegration [9,10] (Figure 11). Surfaces other than mucosal ones were glazed (Optiglaze clear, GC International AG, Luzern, Switzerland), while the mucosal side of the temporaries was polished [11].

Also at this point, the patient was prescribed to take amoxicillin with clavulanic acid in an amount of 875 + 125 mg every 12 h for 7 days. No premedication was given to the patient.

#### 2.2.3. Stage III

Radiological examination (Carestream 8100 3D, Carestream Dental Germany GmbH, Stuttgart, Germany) (Figure 12) and suprastructure dismounting with a torque check were conducted after six months. Analog open-tray impressions from the MUA level were taken (because of the disputed precision of I.O. scans in such cases) together with 360° scans of temporaries, prosthetic field scans and scans of scanflags from the MUA level along with bite registration. Everything was exported to the technical laboratory in coordinates, in order to create 3D-printed try-ins (Figure 13).

During the try-in appointment, the Sheffield passive fit test was performed on the 3D-printed structure with a milled titanium bar. The fit was also crosschecked with long-lasting welded–printed temporaries and the gypsum model [12]. In a CAD software, the gypsum model was fully aligned with scanned models and used to reduce distortions and further material tensions. Any cosmetic changes were implementedwith the use of color flow material in order to demonstrate them to the technical laboratory.

The final step was the delivery of milled ceramic shells on top of an anodized titanium bar [13]. The final prosthesis was made from solid titanium block V grade milled bar combined with a Adite zircon multilayer taco shell using the Imes Icore 350i milling machine. The Luting agent was Panavia v5 Opaque (Kuraray Europe GmbH, Hattersheim, Germany). Final touches to the esthetics were completed using MiYO Kit Color+ (Chemichl AG, Vaduz, Liechtenstein) (Figure 14 and Figure 15). As the fixed temporary restoration was a biocopy of residual dentition and the final restoration was a biocopy of the first biocopy, it was additionally equilibrated stage by stage intraorally with the development of excursion guidance leading to a mutually balanced occlusion in the final restoration. Equilibration was conducted in static and dynamic occlusion with the use of progressive articulation paper and a digital articulation device (OccluSense, Baush Germany GmbH, Hainspitz, Germany). Mutually balanced occlusion was an occlusal design of choice [14,15].

The whole described workflow can be summarized into the following steps:Intraoral scan and CBCT;Meshmixer design of removable dentures;Three-dimensional manufacturing of removable dentures;Extraction of remaining teeth, immediate temporary implant placement, and removable denture delivery.Conduction of DSD, an I.O. scan, a 360° scan of removals, and CBCT after the healing phrase.The merging of the abovementioned examination data and design of the prosthetically driven implant positions.The development of the surgical guide and the 3D-printing of the teeth shells.One-day surgery with implant placement, MUA, and intraoral welding to obtain immediately loaded long-lasting temporaries.The conduction of an analogue impression, an I.O. scan, a removable 360° scan, and a scanflag scan.The conduction of a Sheffield passive fit test of 3D-printed try-ins on a milled titanium bar.Final restoration delivery with occlusal equilibration.

## 3. Discussion

Placing an implant in a suitable 3D position positively influences the success rate and lowers the risk of its future failure [16]. Moreover, digitally designed positions and guided surgery increase the chances of the correct positioning of an implant [17], and even if any deviation occurs, it has no or little clinical significance [18], although in terms of the passiveness of an immediately loaded prefabricated prosthesis, even a minor difference may lead to tensions between the superstructure and implants [10]. This problem has been minimized via intraoral welding and the use of 3D-printed temporary prosthesis, which will be discussed in future research. Regardless, clinicians must understand, verify, and adapt to unexpected events, rather than blindly following the assumed protocol [19,20].

In a study by Cattoni et al. [21], traditional and digital workflows were compared and the results proved that both surgical approaches are valid and comparable in terms of the occurrence of implant failure, differences in marginal bone levels and patient appreciation.

Computer-assisted implant surgery (CAIS) and traditional methods have no statistical difference between them in terms of the length of surgical intervention in cases of partially edentulous patients or single implants, although such a difference was visible in fully edentulous patients [17], which is usually the case in perio-patients.

Similarly to the technique used in this study, the BARI technique shows a full digital workflow, although it starts from edentulous patients using complete dentures [4] rather than patients with periodontitis and residual though hopeless dentition.

In this study, implants were immediately loaded, in order to sustain acceptable esthetic conditions. The passiveness and endurance of prefabricated prosthesis was guaranteed via intraoral welding.

Intraoral welding combines the advantages of a personalized, passive titanium structure, which is bent as needed, according to real, not estimated, implant positions [22,23]. Moreover, the whole structure is rigid enough to diffuse the mastication forces through all implants and lower the risk of implant loss [24].

If the digital workflow is used, it can also be applied to design and manufacture a proper temporary reconstruction.

This study shows the use of different types of prosthetic reconstructions:Those printed on a desktop 3D printer (Phrozen mini 4k, Phrozen Tech. Co., Ltd., Hsinchu City, Taiwan) [6];Those printed and fixed with a rigid welded bar, as a reinforced overdenture shows a lower risk of fracture, when compared to that associated with nonreinforced dentures [25];Those created with a full-porcelain and milled titanium bar (also reinforced).

The possibility of printing with a desktop 3D printer makes it more accessible and faster to design and manufacture. Additionally, it lowers the cost of a single prosthesis and allows the design of a restoration according to DSD [26,27]. Moreover, it enables the creation of low-cost copies, in the case of unfortunate events, among which the fracture of the composite resin of a superstructure is the most common chairside complication [9].

With a properly designed temporary reconstruction, patients can adapt to a new smile and any suggestions or requests may be implemented in the final porcelain restoration.

## 4. Conclusions

This case shows different aspects of digital dentistry: intraoral scanning, designing 3D-printed temporary restorations, the planning and preparation of surgical guides for implant placement, digital smile design (DSD) with adequate occlusion, and scanning for final restoration. As this field of dentistry is in dynamically increasing progress, the next steps may be robotic-assisted surgery, augmented reality, etc.

## Figures and Tables

**Figure 1 reports-06-00052-f001:**
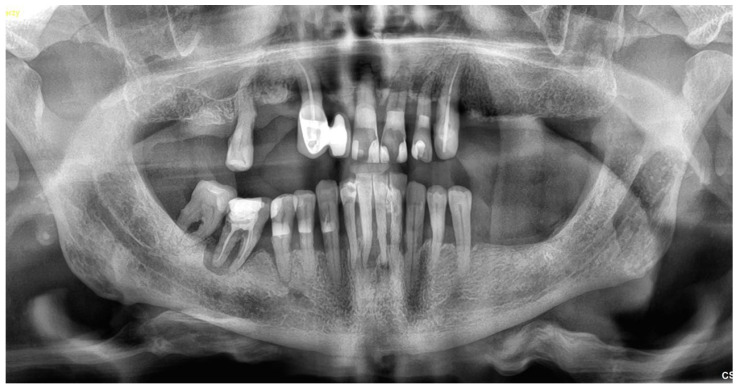
An orthopantomogram (OPG) of baseline dentition.

**Figure 2 reports-06-00052-f002:**
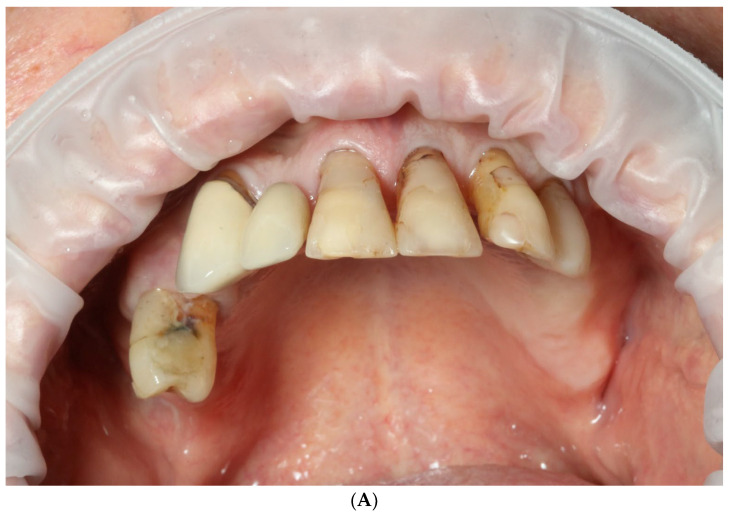
Intraoral view of upper (**A**) and lower jaw (**B**).

**Figure 3 reports-06-00052-f003:**
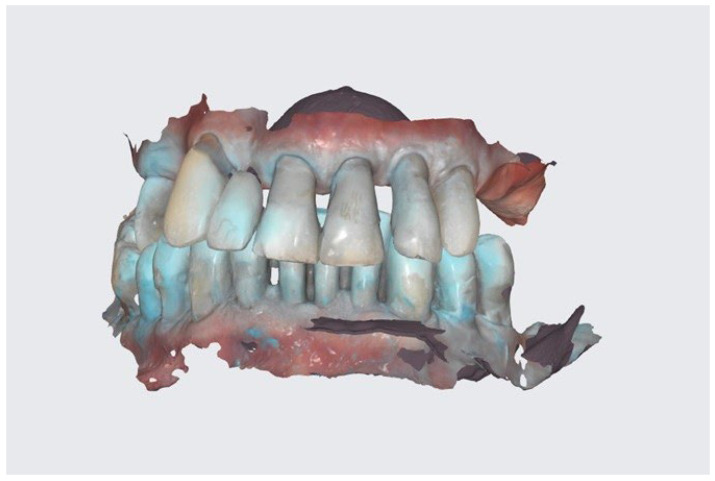
Intraoral scan before treatment.

**Figure 4 reports-06-00052-f004:**
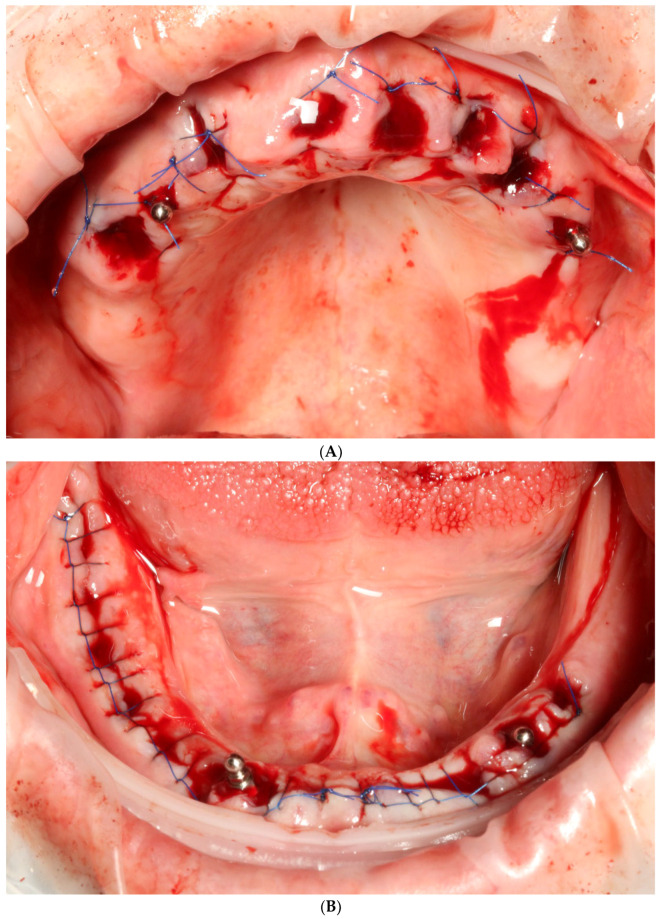
Intraoral view of upper (**A**) and lower jaw (**B**) after extractions and placement of temporary implants for proper retention of printed temporary restorations.

**Figure 5 reports-06-00052-f005:**
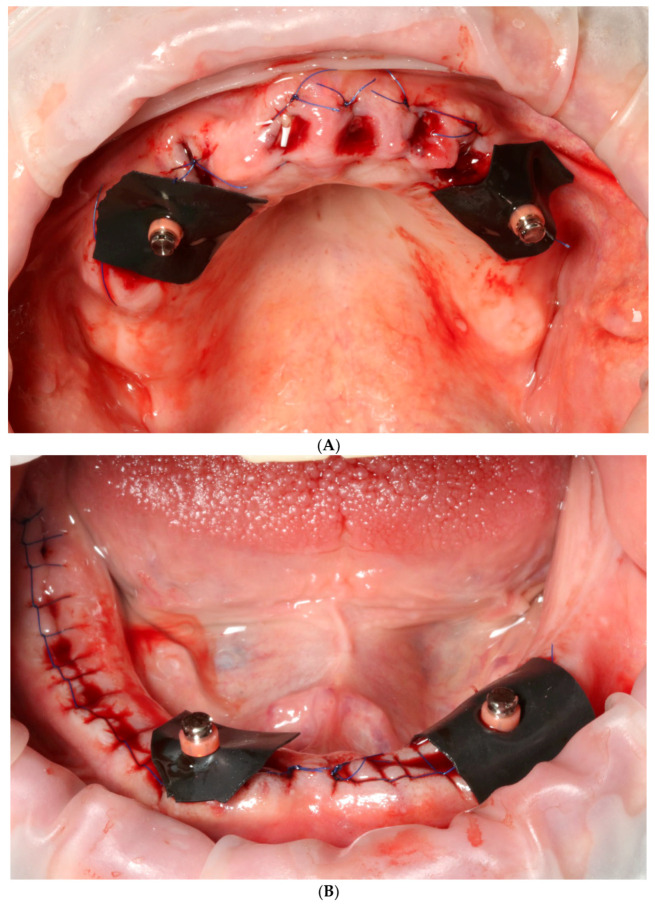
Temporary implant region covered with a rubber piece to prevent the fixing material from entering the wound. Upper (**A**) and lower jaw (**B**).

**Figure 6 reports-06-00052-f006:**
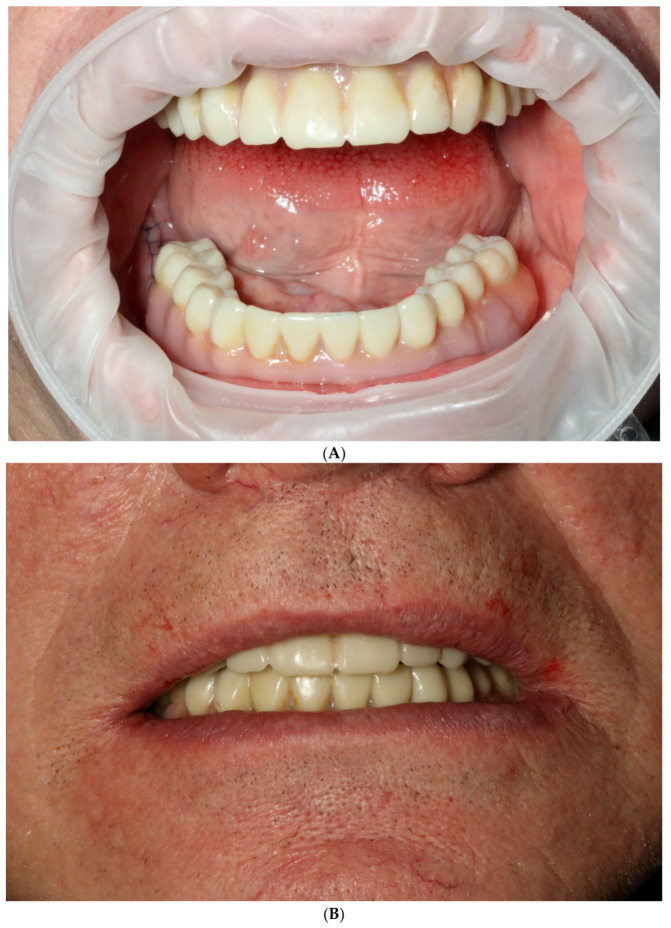
Photographs of 3D-printed temporary restorations; intraoral view (**A**) and extraoral view with a smile (**B**).

**Figure 7 reports-06-00052-f007:**
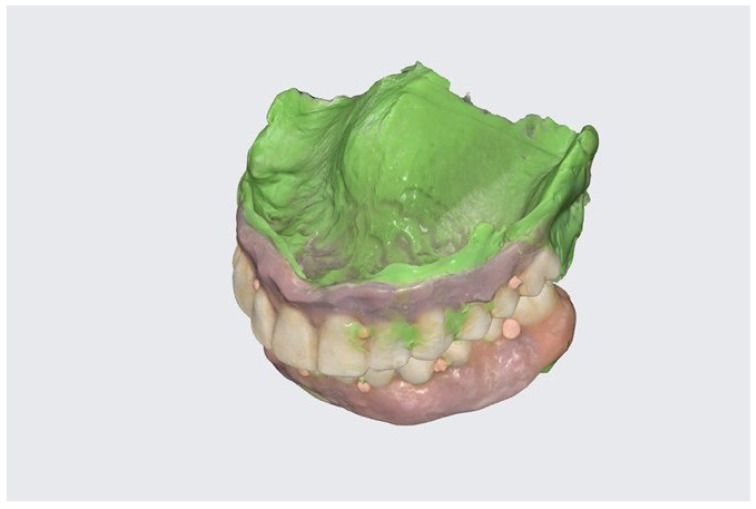
Scan of 3D-printed temporary dentures with gutta-percha radiological tags.

**Figure 8 reports-06-00052-f008:**
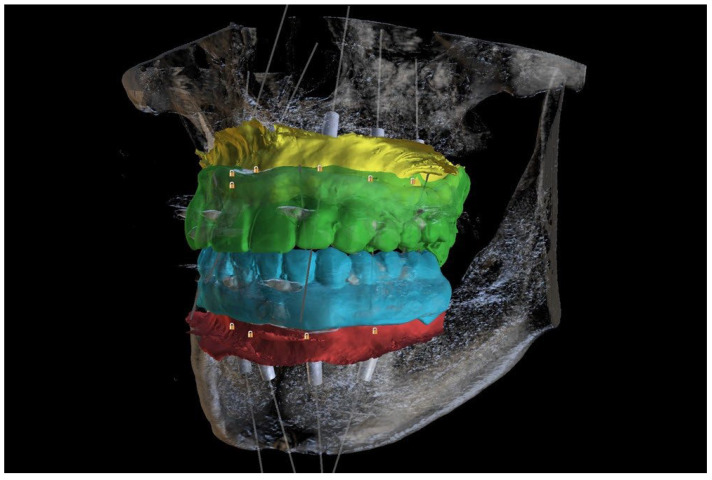
New CBCT, intraoral scans, and a scan of 3D-printed temporary restorations merged together in the Blue Sky Plan program with the prosthetically driven positions of implants.

**Figure 9 reports-06-00052-f009:**
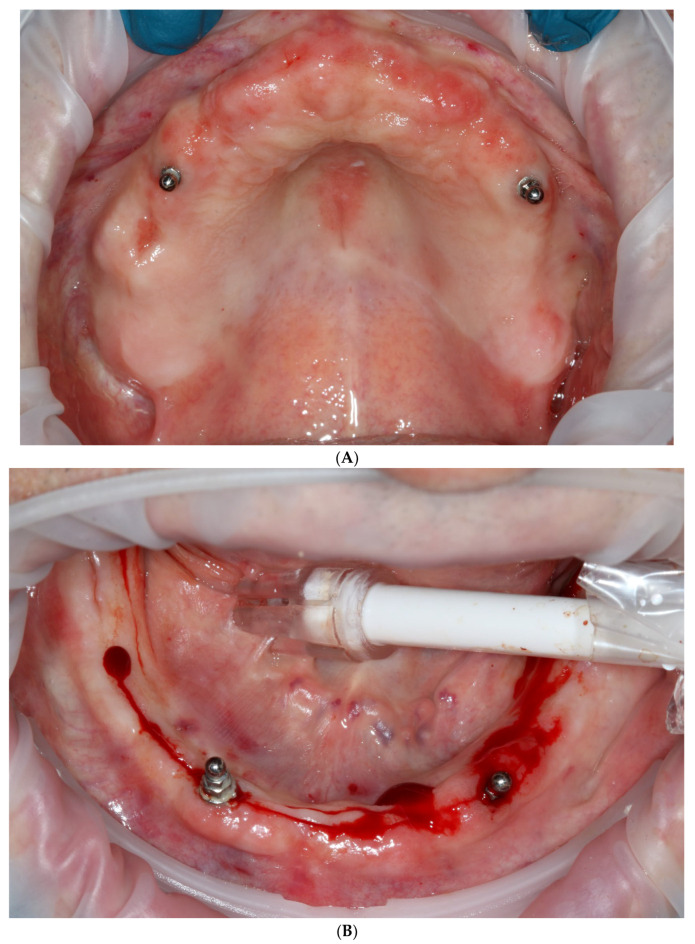
Intraoral view of upper (**A**) and lower jaw (**B**) after 4 months of healing with implant surgical guide (**C**).

**Figure 10 reports-06-00052-f010:**
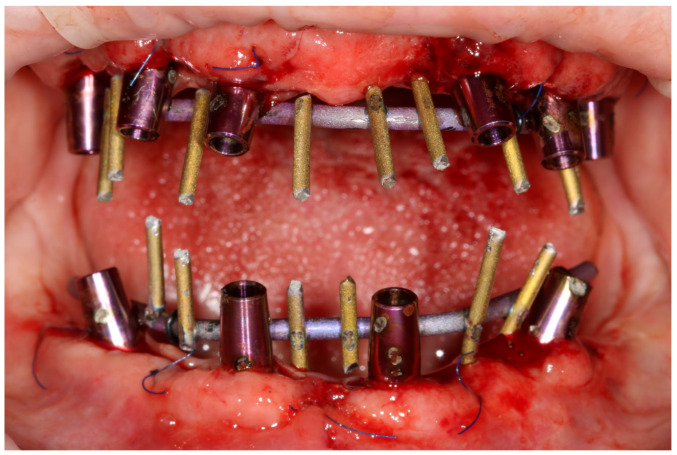
Implants with MUA and AM abutments welded intraorally with 2.0 mm titanium wire allowing the obtention of a rigid framework.

**Figure 11 reports-06-00052-f011:**
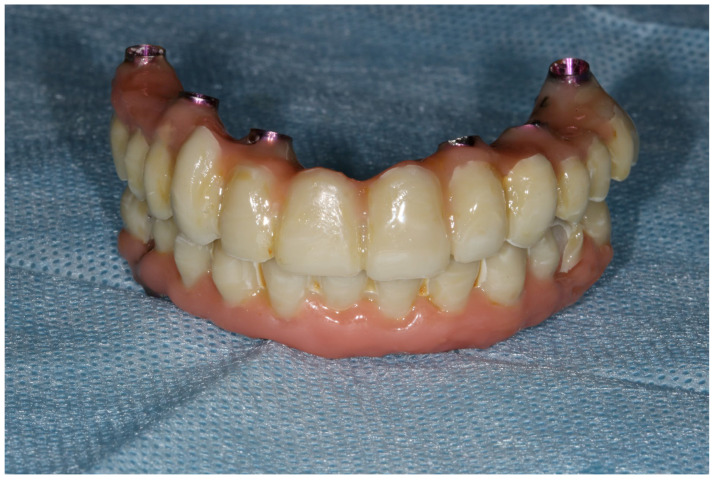
New 3D-printed temporary reconstruction connected with rigid welded framework and MUA as long-term fixed reconstruction for implant integration healing period.

**Figure 12 reports-06-00052-f012:**
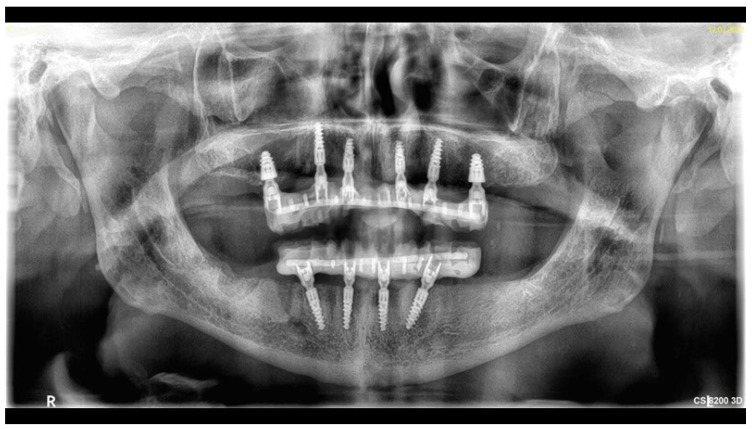
OPG 6 months after placing implants and immediate loading with welded long-lasting temporary reconstruction.

**Figure 13 reports-06-00052-f013:**
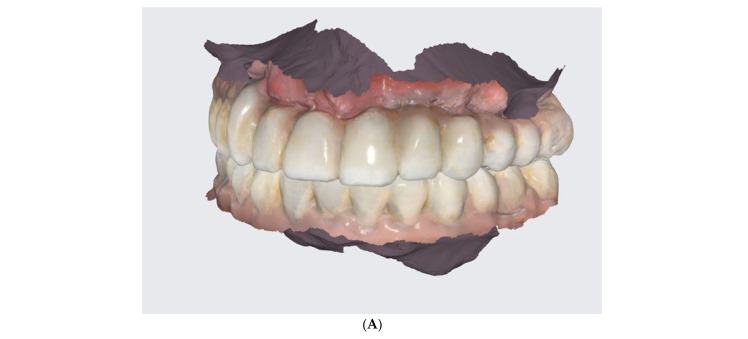
Scans of temporary restorations on MUA (**A**), with scanflags from the MUA level (**B**) and the 3D-printed try-in aligned with the mandible and MUA-level scanflags (**C**).

**Figure 14 reports-06-00052-f014:**
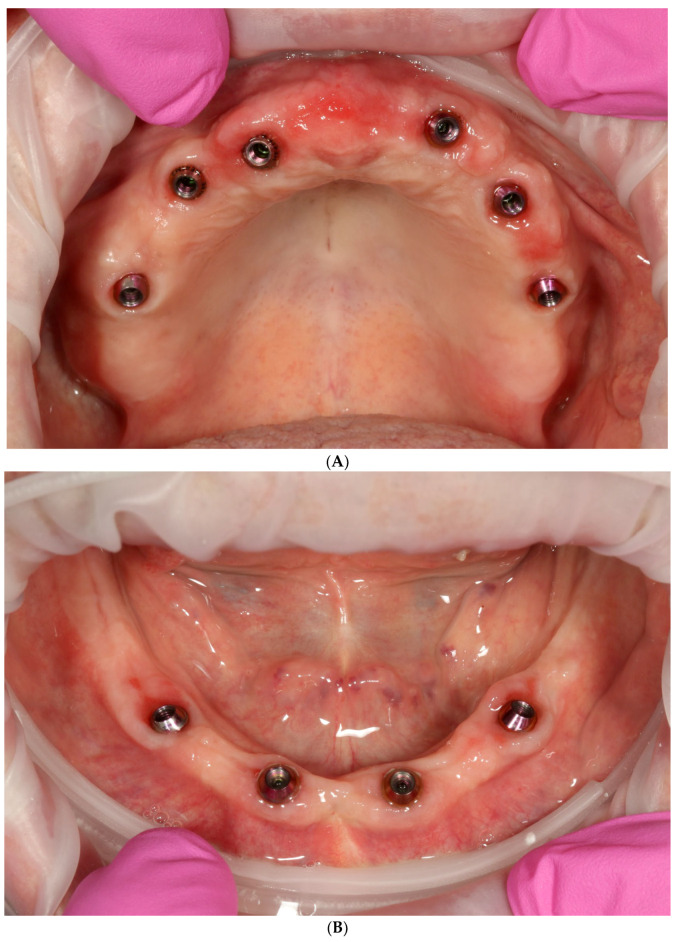
Intraoral view of MUA in upper (**A**) and lower jaw (**B**) before delivery of final restorations.

**Figure 15 reports-06-00052-f015:**
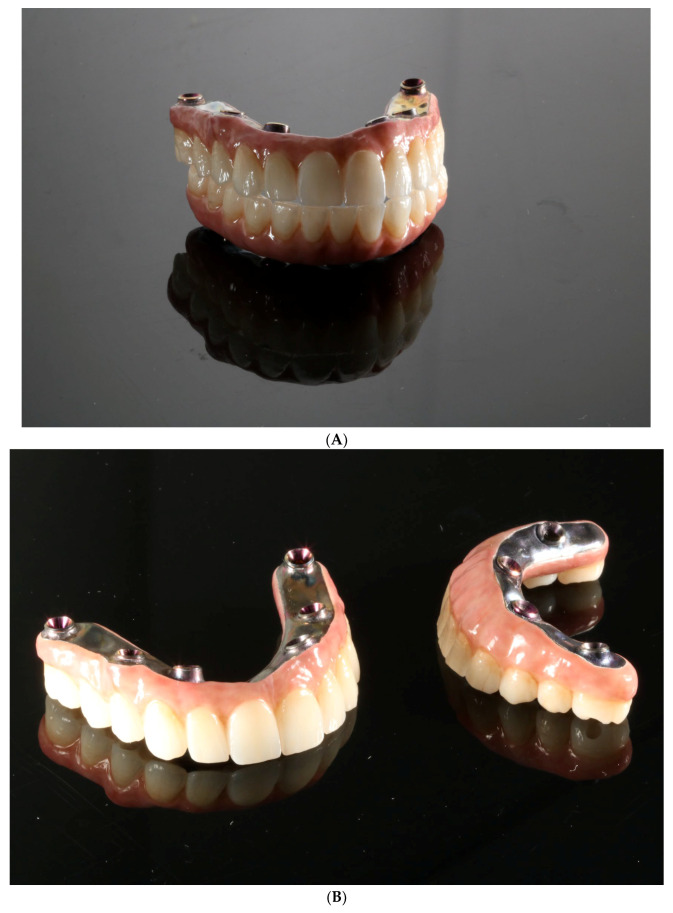
Final, milled, and ceramic restorations on anodized titanium bar before (**A**,**B**) and after delivery (**C**).

## Data Availability

The data presented in this study are available on request from the corresponding author. The data are not publicly available due to privacy isuues.

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
