# Peer review of "Digital Workflow in Full Mouth Rehabilitation with Immediate Loading, Intraoral Welding and 3D-Printed Reconstructions in a Periodontal Patient: A Case Report"

_reports, 2023, doi:10.3390/reports6040052_

Round 1
Reviewer 1 Report
Comments and Suggestions for Authors
The manuscript entitled as “Digital Workflow in Full-Mouth Rehabilitation with Immediate Loading, Intraoral Welding and 3D-printed Reconstructions in a Periodontal Patient - a Case Report” describes a total rehabilitation of a grade 3/stage 4 periodontal patient through implants. The case report has high quality pictures and is well conducted.
The only major concern is about the ethics, which were not addressed by the authors. The patient must consent to the publication of the pictures, even when the patient cannot be recognized. The Helsinki declaration is also mandatory. The reviewer believes that not ethical committee approval is necessary, since is a case report. However, the consent must be addressed.
Below, the authors can find minor concerns and suggestions.
Introduction:
Succinct and straightforward, exposing where the digital workflow can be useful. Good job. As the only suggestion, change treatment for extensive rehabilitations, line 27.
Patient information:
Figure 1. Add the meaning of the abbreviation (OPG).
As a suggestion, write Stage I in instead of I stage. Apply for the subsequent sections.
Line 54: periodontal diseases are currently classified in grades and stage. Correct the stadium term.
The authors explained the reason for the rejection of immediate implantation, as well as edentulism. Then, the authors propose the alternative, that is, immediate removable dentures over temporary implants. As a suggestion, add the temporary implant insertion in line 56.
The health status of the patient could be added as well (e.g., medications, chronic pathologies).
The reviewer suggests the addition more information about the surgery, as flap/flapless access, the temporary implants, that is, if they were inserted using a micromotor/hand piece or manually, the number of screws and their position, suture, post operative therapeutics.
Figure 8: To add more information, the authors could place small cuts of the tomographic region, with the draw of the digital implant, to illustrate the rationale of the surgical guide.
As in the placement of the temporary implants, the authors could add more information about the flap/flapless access and the removal of the temporary implants. Moreover, the attachment of the rigid frameworks to the 3D hollow printed shells (line 143) should be described.
Discussion:
Properly conducted. As a suggestion, discuss the removal of molding and plaster models can help to reduce distortions, and therefore, tensions.
Comments on the Quality of English LanguageMinor editing of English language required.
Reviewer 2 Report
Comments and Suggestions for Authors
See changes on pdf

Comments on the Quality of English Languagelow
Round 2
Reviewer 1 Report
Comments and Suggestions for Authors
The reviewer congratulates the authors for the revised version of the manuscript. The surgical steps and implant description enhanced the readability and the discussion of the case report. Authors addressed the ethical concerns.
Congratulations.
Comments on the Quality of English LanguageMinor editing of English language may be necessary.
Reviewer 2 Report
Comments and Suggestions for Authors
Fine.